# Relationship between Cytotoxicity and Surface Oxidation of Artificial Black Carbon

**DOI:** 10.3390/nano11061455

**Published:** 2021-05-31

**Authors:** Yen Thi-Hoang Le, Jong-Sang Youn, Hi-Gyu Moon, Xin-Yu Chen, Dong-Im Kim, Hyun-Wook Cho, Kyu-Hong Lee, Ki-Joon Jeon

**Affiliations:** 1Department of Environmental Engineering, Inha University, Incheon 22212, Korea; lethihoangyen@inha.edu (Y.T.-H.L.); hyunwook@inha.ac.kr (H.-W.C.); 2Program in Environmental and Polymer Engineering, Inha University, Incheon 22212, Korea; 3Department of Energy and Environmental Engineering, The Catholic University of Korea, 43 Jibong-ro, Bucheon-si 14662, Korea; jsyoun@catholic.ac.kr; 4Jeonbuk Department of Inhalation Research, Korea Institute of Toxicology, Jeongeup 53212, Korea; higyu.moon@kitox.re.kr (H.-G.M.); chen.xinyu@kitox.re.kr (X.-Y.C.); Dongim.kim@kitox.re.kr (D.-I.K.); 5Department of Human and Environmental Toxicology, University of Science & Technology, Daejeon 34113, Korea

**Keywords:** artificial black carbon (aBC), thermal treatment, cytotoxicity, reactive oxygen species (ROS)

## Abstract

The lacking of laboratory black carbon (BC) samples have long challenged the corresponding toxicological research; furthermore, the toxicity tests of engineered carbon nanoparticles were unable to reflect atmospheric BC. As a simplified approach, we have synthesized artificial BC (aBC) for the purpose of representing atmospheric BC. Surface chemical properties of aBC were controlled by thermal treatment, without transforming its physical characteristics; thus, we were able to examine the toxicological effects on A549 human lung cells arising from aBC with varying oxidation surface properties. X-ray photoelectron spectroscopy, as well as Raman and Fourier transform infrared spectroscopy, verified the presence of increased amounts of oxygenated functional groups on the surface of thermally-treated aBC, indicating aBC oxidization at elevated temperatures; aBC with increased oxygen functional group content displayed increased toxicity to A549 cells, specifically by decreasing cell viability to 45% and elevating reactive oxygen species levels up to 294% for samples treated at 800 °C.

## 1. Introduction

Black carbon (BC) is an undesired byproduct from the incomplete combustion of fossil fuels and biomass [1,2,3,4,5]. BC, commonly referring to soot and found in fine particulate matter (PM_2.5_), is the main component of atmospheric carbonaceous aerosols [1,2,3]. As an efficient light-absorbing carbonaceous material, BC has been mainly implicated as a short-lived climate forcer [4,5,6]. Besides its impact on climate change and the ecosystem, BC has also been associated with pulmonary, cardiovascular, and premature death [3,6,7].

BC is characterized as having fractal agglomerates, being insoluble in water or common organic solvents, and being refractory and potential toxic [2,8]. The factor of human health was less concerned with than climate change, however, a number of studies focus on health impact assessments, which have usually been conducted by estimating the related adverse health outcomes of the population based on the concentration of BC [3,9]. Due to the lack of laboratory BC samples and the inability to separate BC from atmospheric particulate matter, to date, no specific toxicity experiment induced by BC has been conducted [10].

On the other hand, engineered carbon nanoparticles with corresponding toxicity were widely carried out [11,12]. Carbon nanoparticles have been shown to induce inflammation, enhance oxidative stress, and transform cell signaling and gene expression in mammalian cells and organs, and toxic effects arising from their well-defined physical features have been described in numerous studies [13,14,15,16]. Cheng et al. tested ultrafine carbon nanopowder and pointed that carbon particles disrupted the keratinocyte differentiation and upregulated inflammation; carbon powder was claimed to mimic ambient ultrafine particles, however, the commercial carbon powder properties are far different from real-world atmospheric particles [17]. Engineered carbon black and BC share some similar features (black appearance, aggregate morphology, and elemental component) and even some common biological responses; therefore, some of the toxicology results have misunderstood carbon black interchangeably with BC [10,18]. Hong et al. [18], based on elemental analysis, claimed that engineered carbon black with a high, pure percentage of elemental carbon is different from BC, therefore, an experiment with BC that is regarded to have intensive toxicity is needed.

The diesel soot particulate matter is produced by the National Institute of Standards and Technology (NIST), which can mimic real-world diesel soot, however, the existence of polycyclic aromatic hydrocarbon (PAH) with other chemical compositions renders the identification of specific parameters responsible for inducing toxicity and the mechanisms by which they cause harm to the human body highly challenging [19]. Moreover, soot samples were also collected from several sources with a complicated mixture in which BC was barely separated [20,21,22]. With a mixture of inorganic compounds, trace metals, and PAH, it is challenging to determine the key factors of soot that play the main roles in inducing toxicity. Therefore, artificial BC (aBC) is essential for a toxicity test satisfying three principal requirements: (1) representing elementary atmospheric BC; (2) maintaining origin physical properties; and (3) having a surface that is chemically controllable.

Herein, we have synthesized aBC using an aerosol generator. We defined aBC in this study as the ultrafine particles generated from one origin source, graphite, to simplify the test of relation between toxicity and the chemical surface of aBC; aBC must possess a controllable chemical surface and common physical properties. Thermal treatment is a straightforward method for manipulating the chemical properties of aBC surfaces, especially to control oxygen functional group content while maintaining its physical characteristics. Because a dominant cytotoxicity mechanism of ultrafine particles is a cell–particle interaction accompanied by the overproduction of ROS [23,24,25], ROS generation and related cytotoxicity were the focus of this study. We demonstrated that an increase in oxygenated functional groups on the surface of BC triggered increased cytotoxicity and ROS levels in the human lung cancer cell line (A549).

## 2. Materials and Methods

### 2.1. Particle Generation and Thermal Treatment

Graphite was selected to produce aBC generated from a spark discharge soot generator (DNP Digital 3000, Palas GmbH, Karlsruhe, Germany). It operates on the principle of spark discharge to produce nanoscale particles of consistent concentration in high yields and negligible volatile content. The generator maintains a jump spark between two graphite electrodes at a high voltage of 2500 V. Graphite is vaporized by this spark, and then condenses to form particles with a size distribution and structure similar to those of diesel soot. The generator was operated under a constant flow of carrier inert Ar gas and purified dilution air at flow rates of 5 and 10 LPM, respectively (gas streams were controlled with a series of mass flow controllers).

The generated particles were produced in a tubular furnace at different ranges of temperatures. The first experiment was performed with the furnace temperature increasing continuously from room temperature (RT) to 800 °C to investigate particle properties as a function of the treatment temperature. Five treatment temperatures (RT, 200 °C, 400 °C, 600 °C and 800 °C) were applied separately in the second experiment to evaluate the effects of temperature on the physicochemical properties of aBC. A schematic of the experimental setup, including the sampling and measuring devices, is presented in Figure 1.

### 2.2. Physicochemical Characterization

An electrical low-pressure impactor (ELPI+^TM^, DEKATI, Finland) was employed to measure the concentration of generated particles in the range of 6 nm to 10 μm with a time resolution of 1 s. Furthermore, we estimated the surface area concentration of the deposited particles in the human respiratory tract based on ELPI+ stage-specific conversion factors. Teemu et al. estimated the ELPI+ response coefficient (β) by using an equivalent unit density (1 g/cm^3^) for spherical particles, with a possible error in the mean diameter arising at each stage of each instrument 2% [26]. From the measured electrical current (I) carried by the aBC, the deposited area concentration of aBC (A_dep_) was estimated for three regions, head airways, tracheobronchial, and alveolar, according to the following equation:Adep=β × I

The deposited area distribution of aBC was estimated to select suitable cell lines for cytotoxicity testing.

In the second experiment, samples were collected on pre-fired quartz fiber filters (six hours, 600 °C) with a diameter of 25 mm (FTQ25, Zefon, Ocala, FL, USA) and Teflon filters (R2PJ047, Pall Corporation, Port Washington, NY, USA) with a diameter of 47 mm using filter holders at five different temperatures (RT, 200 °C, 400 °C, 600 °C and 800 °C); these samples were utilized for chemical analysis and the in vitro test.

High-resolution transmission electron microscopy (HR-TEM; JEM-2100, JEOL, Akishima, Japan) was employed to observe the size and morphology of the aBC at an accelerating voltage of 200 kV. The aBC was collected on holey carbon TEM grids using a mini particle sampler (MPS, Ineris, Ecomesure, Saclay, France). For SEM analysis, aBC was dispersed on a heated (100 °C) silicon wafer by depositing a drop of particle suspension, created by sonicating aBC collected on a quartz filter in DI water. These samples were coated with a thin film of Pt and then observed by field emission-scanning electron microscopy (FE-SEM; S-4300, Hitachi, Tokyo, Japan). 

The surface chemical compositions of the aBC samples were analyzed by X-ray photoelectron spectroscopy (XPS; Thermo Fisher Scientific Co., Walthem, MA, USA). Additionally, Raman spectra were acquired using an Xplora spectrometer (Horiba Jobin-Yvon, Palaiseau, France) equipped with a 532 nm solid state laser source. Fourier transform infrared vacuum spectroscopy (FTIR; Bruker VERTEX 80V, Coventry, UK) was employed to investigate structural changes occurring during the heating process. 

### 2.3. Endocytosis, Cytotoxicity Assay, and Evaluation of Reactive Oxygen Species (ROS)

#### 2.3.1. Cell Culture 

A549 cells were purchased from the American Type Culture Collection (ATCC, Manassas, VA, USA). The cells were cultured in Roswell Park Memorial Institute (RABCI) 1640 (Gibco, Grand Island, NY, USA) containing 5% fetal bovine serum (FBS; Gibco) and 1% streptomycin and penicillin at 37 °C in an atmosphere containing 5% CO_2_.

#### 2.3.2. Cytotoxicity

For cytotoxicity experiments, A549 cells were seeded in 96-well plates at a density of 1 × 10^4^ cells per well and left to attach overnight. The cells were then treated with various concentrations (50–2000 µg mL^−1^) of aBC in the complete medium for 48 h. This was followed by the addition of 3-(4,5-dimethylthiazol-2-thiazyl)-2,5-diphenyl-tetrazolium bromide (MTT; Sigma Aldrich, St. Louis, MO, USA) solution to the final concentration of 1.0 mg mL^−1^. After 3 h of incubation, the medium was removed, the formed blue formazan crystals were dissolved in 100 μL of dimethyl sulfoxide (Junsei Chemical Co., Tokyo, Japan) per well DMSO, and the absorbance at 570 nm (690 nm background control) was measured using a microplate reader (Biotek, Winooski, VT, USA). Low toxicity of aBC was observed in most treatments, with 75% of cells remaining viable after treatment with the highest dose (2.0 mg mL^−1^) of aBC for 48 h. Therefore, these treatment conditions were used for all subsequent experiments. 

#### 2.3.3. Endocytosis of aBC in A549 Cells

The aBC-exposed cells were harvested using 0.25% trypsin-EDTA (Gibco, Thermo Fisher Scientific, Walthem, MA, USA) from culture plates and cyto-centrifuged (800 raBC; 10 min) onto glass microscope slides using Cytospin 4 (Thermo Fisher Scientific, Walthem, MA, USA). Cell smears were fixed in methanol for 1 min and stained using Diff-Quik solution (Dade Diagnostics, Aguada, Puerto Rico). Stained cells were analyzed under a light microscope (Leica, Wetzlar, Germany).

#### 2.3.4. Measurement of Reactive Oxygen Species (ROS) Levels

A549 cells were incubated for 30 min at 37 °C in RABCI 1640 medium containing 3.3 μmol L^−1^ of 2,7-dichloro-fluorescein diacetate (DCF-DA) (Thermo Fisher Scientific, Walthem, MA, USA) for ROS detection. The cells were treated with aBC (2.0 mg mL^−1^ for 48 h), then washed with Dulbecco’s phosphate-buffered saline (DPBS; Welgene Inc., Daegu, Korea). The DCF-DA intensity in the cells was immediately measured at 495 nm (excitation)/529 nm (emission) using a microplate reader. ROS production in the cells is represented as a percentage of DCF-DA intensity relative to cell viability in each well, which was defined as 100%.

## 3. Results

### 3.1. Emission Characteristics of Synthesized aBC

Figure 2 indicates that the generated particles experienced a change in the number concentration during thermal treatment. The left vertical axis represents the aerodynamic diameter of aBC, and the contour map reflects a reduction in the particle number concentration going from red to blue. This result demonstrated that the total particle number concentration was at a maximum of 1.6 × 10^8^ cm^−3^, decreasing with higher treatment temperatures and eventually reaching 2.2 × 10^7^ cm^−3^ when the furnace temperature was 800 °C. The size distribution is also provided here, with the majority of the produced particles exhibiting aerodynamic diameters smaller than 100 nm. The aerodynamic diameter has been reported as the dictation of particle penetration into the lung, therefore, with those nano-scaled sizes, aBC has a higher chance to travel deep into the human lung, even meeting the bloodstream, cells, and tissues [27,28]. All size distributions were found to be unimodal, with a typical size of approximately 30 nm in aerodynamic diameter. The dominant size of the particles increased marginally at approximately 400 °C and 800 °C. The slight increase in particle size and decrease in the number concentration at higher temperatures were likely caused by thermal coagulation. Higher temperatures enhance attractive forces and Brownian motion, resulting in an increased frequency of collisions [29]. At every collision, there is one less particle, and no new growth or nucleation occurs during aggregation, leading to a small increase in size, and the thermal degradation of particles chiefly decreases the number concentration [30].

### 3.2. TEM and SEM Image Analysis

TEM and SEM images of the particle samples were collected for the five treatment temperatures, as shown in Figure 3. SEM images indicated particle super-aggregation, with a tendency for extremely small particles to assemble into bigger particles. The majority of the aBC was approximately 27.7 ± 3.8 nm in diameter, with a near-spherical morphology (Figure 3a). In Figure 3b, the TEM results provide detailed images of the particles. Particles aggregated into branching structures and presented typical diesel soot morphologies, with nearly spherical and irregular carbonaceous particles. The primary particle size distribution is presented in Figure 3c, with a dominant size of approximately 30 nm, which is comparable to the real-time size distribution data shown in Figure 2. The general size and shape of the synthesized aBC revealed a uniform, near-spherical shape with a narrow size distribution. TEM and SEM images indicated that varying the treatment temperature did not result in significant differences in particle morphology. 

### 3.3. Chemical Surface Properties

The XPS element and atomic concentration survey (Figure 4a) showed that the presence of Si2p is the highest from the blank sample due to utilization of a quartz filter. The concentration fraction of O1s slightly went up in 600 °C samples, and significantly increased in the 800 °C sample compared to others, indicating that the oxygen increased. The XPS carbon (C1s) spectra of the five samples are depicted in Figure 4b, and are de-convoluted into four peaks. The C–C bond, or graphitic carbon, was dominant, represented by the green peak at approximately 284.2 eV [31,32,33]. Other peaks at higher binding energies were related to carbon-oxygenated functional groups, including: C–O at 285.6 eV (red peak), –C–OH at 286.8 eV (pink peak), and –C=O at 288.7 eV (blue peak) [30,31,32]. The component fractions were assessed as functions of furnace temperature in Figure 4c. The C–C content decreased from 66% to 59%, while C–O increased from 16.4% to 19%, and C=O increased from 5.5% to 9.5% with increasing treatment temperature from RT to 800 °C. This change indicated that the percentage of oxygen functional groups had increased, resulting from higher amounts of oxygen present on the sample surfaces at higher temperatures. We assumed that the oxidation of graphitic carbon occurred above 600 °C.

To verify the oxidation of aBC, vibrational characterization was performed by Raman and FTIR analysis. Two typical overlapping peaks are visible in the Raman spectra displayed in Figure 5a, namely a D peak at 1340 cm^−1^ and a G peak at 1600 cm^−1^ [34,35,36,37]. The D band represents in-plane breathing vibrations of the aromatic ring structures (A_1g_ symmetry), and the G band is the in-plane stretching vibration of sp^2^ carbons (E_2g_ symmetry) [38,39]. The intensity ratios between the D and G bands (I(D)/I(G)) were 0.80, 0.82, 0.82, 0.84, and 0.96 at RT, 200 °C, 400 °C, 600 °C, and 800 °C, respectively. The increase in the I(D)/I(G) ratio at 800 °C was ascribed to an increase in the in-plane breathing vibrations of the aromatic ring resulting from the appearance of a functionalized group on the aromatic ring. This result provides conclusive evidence of aBC oxidation by thermal processing, especially at 800 °C.

Figure 5b shows the FTIR spectra of the five analyzed samples. No significant changes were observed among the spectra of the three samples treated in the range of RT to 400 °C. These spectra presented a sharp peak at approximately 1635 cm^−1^, assigned to aromatic C=C. Three vibrations located at 2852, 2922, and 2962 cm^−1^ were assigned to asymmetric and symmetric C-H stretching of CH_3_ and CH_2_ aliphatic groups [40,41]. At 600 °C, the spectrum included an additional carbonyl C=O stretching (1720 cm^−1^) shoulder, becoming a notable peak at 800 °C and indicating the presence of oxygen functionalities [41,42]. 

### 3.4. In Vitro Toxicity of aBC

Exposure to aBC results in harmful effects on human health, causing pulmonary and cardiovascular diseases [43,44]. The aBC is internalized by various immune and structural cell types, such as macrophages, lymphocytes, skin keratinocytes, and epithelial cells [45,46,47]. We deduced that aBC would be predominantly deposited in the alveolar region of the human lung by analyzing the electrical current carried by aBC (Figure 6). The estimation of the lung deposited surface area distribution based on the current charge carried by aBC implied that a higher fraction (50%) of aBC is possibly deposited in the alveolar region than on other regions of the human respiratory system. Alveolar macrophages and alveolar epithelial cells are highly likely to be the primary and secondary targets for aBC if inhaled by humans. 

Herein, we selected the A549 human lung alveolar basal epithelial cell line for in vitro testing. To verify whether aBC directly affects epithelial cells, we confirmed endocytosis of aBC in A549 cells. Diff-Quik staining indicated the appearance of aBC in the cytoplasm and in the region near the nucleus (Figure 7), indicating the appearance of aBC in the cytomorphologic evaluation; however, to confirm the cellular uptake of aBC, further investigation is needed. The particle size is known to have significant effects on the interactions between nanoparticles and the cellular environment [13]. Here, nano-sized aBC has a greater opportunity for cellular uptake via endocytosis by membrane wrapping due to its effective binding to membrane receptors. When aBC interacts and penetrates through the membrane, the defense mechanism is activated, and cell damage occurs [48,49]. In addition, to determine the cytotoxicity of aBC, we evaluated cell viability using the MTT assay. Cytotoxic effects of the particles were clearly observed in epithelial cells 48 h after stimulation with aBC. The results revealed that aBC exhibited considerable cytotoxicity to A549 cells at a concentration of 2 mg/mL, reducing the survival rate of A549 cells to 75%, 68.2%, 68.5%, and 65% when synthesized at RT, 200 °C, 400 °C, and 600 °C, respectively. In particular, the 800 °C sample induced the strongest cytotoxic effect (cell viability was reduced up to 45%) compared to the naïve control (Figure 8a). Hence, aBC had adverse effects on the survival of A549 cells; aBC directly contacts the cell membrane, inducing membrane stress by disrupting and damaging it, resulting in cell death. Even exposed to a high dose of aBC, the decreased cell viability was still low compared to other literature due to the aggregation of aBC in the cell medium [11,50].

Exposure to aBC induces cell damage via oxidative stress mediated by ROS [47,51]. Excess ROS levels trigger oxidative stress that disrupts cellular homeostasis and affects the oxidation of biomolecules, including DNA, lipids, and proteins [44,52]. ROS production caused by aBC in A549 cells was evaluated by measuring the fluorescence intensity of DCF-CytDA. The aBC generated at higher temperatures induced significant changes in ROS levels in A549 cells compared to the naïve controls (Figure 8b). As observed for cytotoxicity, the greatest oxidative stress was demonstrated in the 800 °C aBC samples for particle-stimulated A549 cells. These results indicated that synthesized aBC directly induced ROS-mediated human alveolar basal epithelial cell damage. The production of high levels of ROS causes significant damage to DNA, thereby affecting cell survival. The correlation of biomass burning aerosol chemical composition with ROS production was used for investigation, however, due to the complex effect of chemical compositions, soot-induced bio-toxicity with ROS production was barely interpreted. It is claimed that no significant correlation can be figured out in the ROS response with measured chemical compositions, and the aging process of biomass burning samples even drove the toxicity test to be more complicated [53]. Therefore, to minimize the factors on real-world soot, with the important roles played in the bio-toxicity results, we highlighted the ROS response with the change of the chemical surface on controlled aBC. 

The physical characterization illustrated a uniform size and shape of aBC synthesized under varying treatment temperatures. On the other hand, Raman and FTIR analysis supported by XPS data led to the conclusion that a greater number of oxygenated functional groups were present on the surface of aBC treated at elevated temperatures. Cell viability and ROS level results indicated that cytotoxicity increased with increasing oxygenated functional group content on the surface of aBC. This agrees with the findings of Das et al., who investigated the correlation between cellular toxicity and oxygenated functional group density on graphene oxide (GO) [48]. They proved that the presence of organic functional groups on the surface of GO affected its interaction with mammalian cells at the “nano–bio” interface, and that an increase in oxygen functional groups rendered GO less biologically inert and resulted in elevated cytotoxicity. We can therefore infer from the present results that an increase in oxygen functional groups on the surface of aBC activates the “nano–bio” interface, thereby facilitating cell membrane disruption by aBC and resulting in higher cytotoxicity. Furthermore, oxidized flame soot increases ROS levels, as reported by A. Holder. This suggests that the increased cytotoxicity of oxidized soot is due to its ability to generate oxidants [50]. Consequently, studies centered on the evaluation of surface chemical properties associated with cytotoxicity suggest that oxygenated functional groups present on the surface of aBC and aBC oxidation are directly related to cell death and oxidative stress.

## 4. Conclusions

The aBC was successfully generated at various treatment temperatures without transforming the original physical features. The treatment temperature is known to significantly impact the surface chemical structure of aBC; hence, we were able to simplify the investigation into the effect of surface chemical properties of aBC on human epithelial cells; aBC with a nano size effectively penetrated the plasma membrane, leading to cell damage. It was found that an increase in the presence of oxygen functional groups on the surface of aBC directly affected cell viability and oxidative stress in A549 cells. The remaining limitation of this research is that aBC is still far from real-world BC, which has a complex mixture of chemical components, however, the results of the relationship between cytotoxicity and surface oxidation of aBC can be the foundation to conduct further research. Furthermore, aggregation and surface morphological changes of aBC in cell medium culture need further investigation, and aBC-induced bio-toxicity with proof of cell uptake is also needed to improve the quality of the cytotoxicity test. The semi-quantification of cell uptake should be investigated to support the mechanism of cell death. An in vivo test and in vitro test with an air–liquid interface approach would be needed for further toxicological models for our synthesized aBC, rather than a submerged cell culture to minimize the current limitations. Future studies will focus on aBC synthesized with sulfates, nitrates, metals, and organic compounds that are commonly emitted with BC to the atmosphere to better simulate real-world experimentation. 

## Figures and Tables

**Figure 1 nanomaterials-11-01455-f001:**
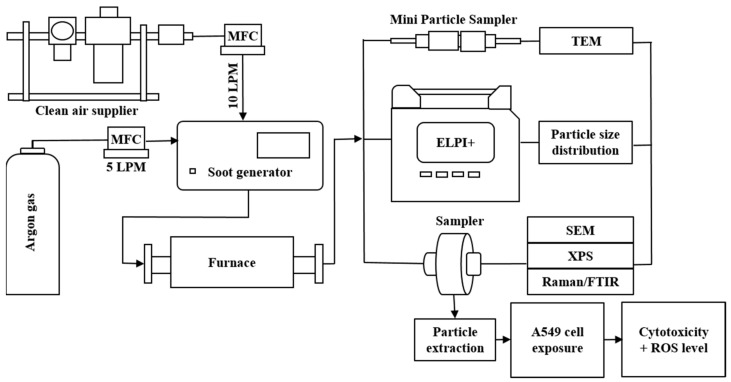
Schematic of the experimental setup.

**Figure 2 nanomaterials-11-01455-f002:**
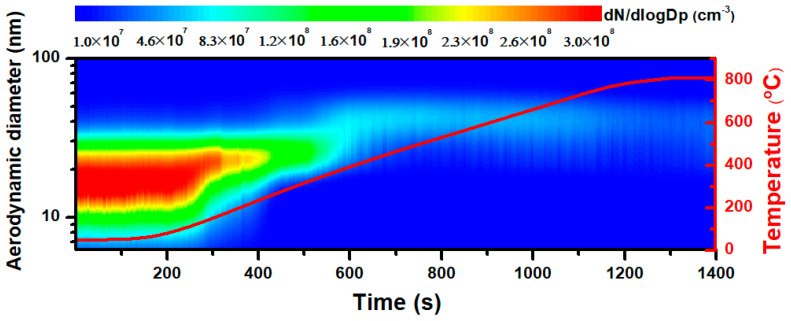
Size and time-resolved particle number concentration during heating from room temperature (RT) to 800 °C. The red line represents the furnace temperature.

**Figure 3 nanomaterials-11-01455-f003:**
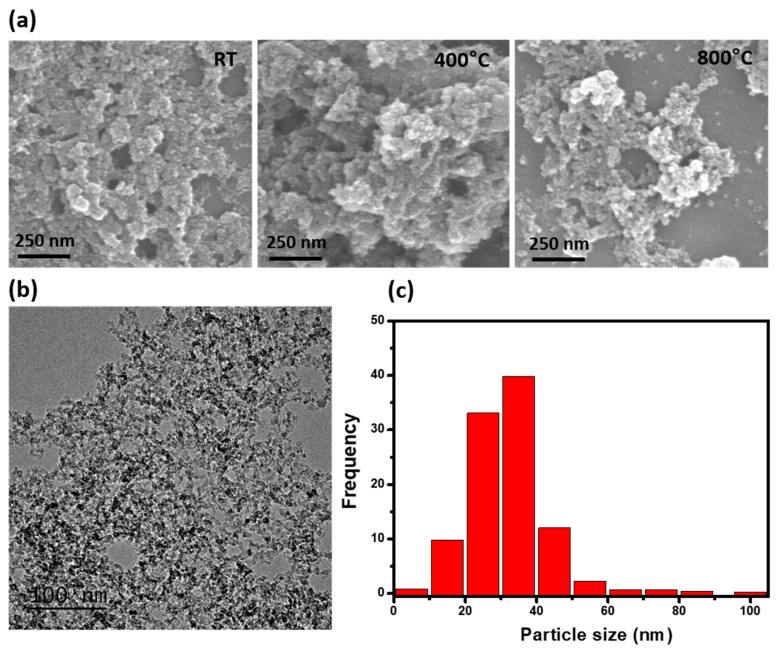
(**a**) SEM and (**b**) TEM images of aBC generated at various treatment temperatures, and (**c**) average particle size distribution for synthesized aBC.

**Figure 4 nanomaterials-11-01455-f004:**
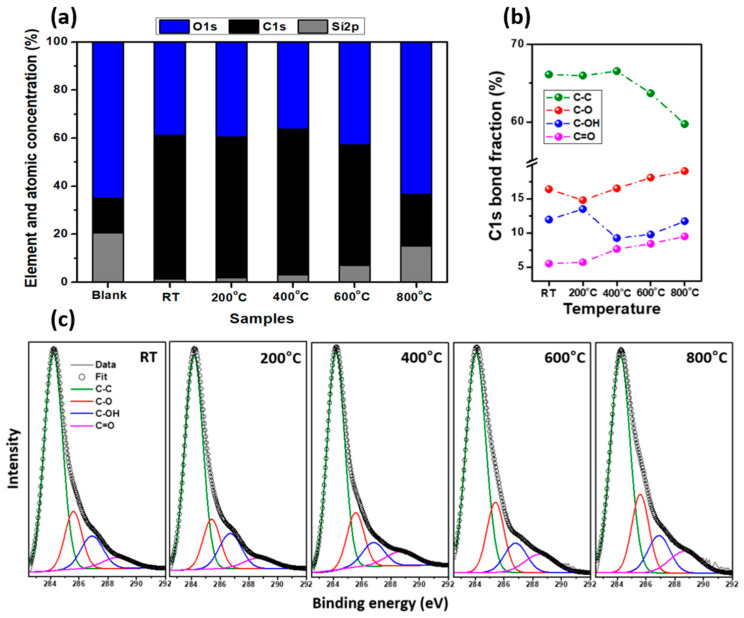
(**a**) Element and atomic concentration survey; (**b**) the C1s bond fraction indicates the variation in C–C, C–O, C–OH, and C=O content with increasing treatment; and (**c**) XPS spectra of synthesized aBC samples treated at five different temperatures.

**Figure 5 nanomaterials-11-01455-f005:**
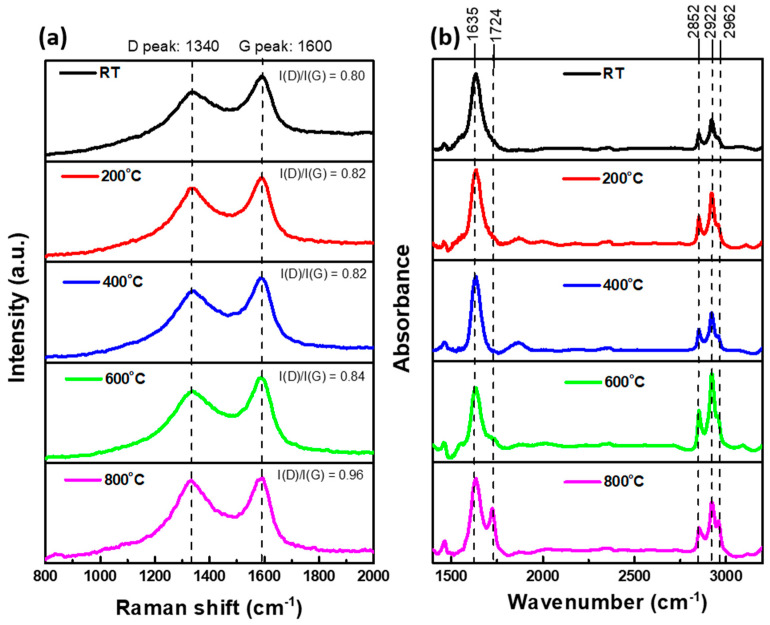
(**a**) Raman and (**b**) FTIR spectra obtained for five aBC samples generated at specific treatment temperatures.

**Figure 6 nanomaterials-11-01455-f006:**
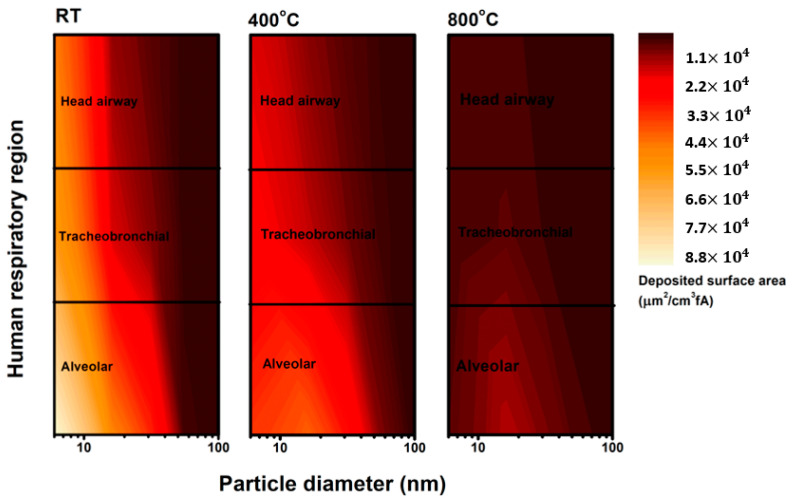
Deposited surface area distributions of aBC in three regions of the human respiratory tract, as predicted based on the electrical current recorded by ELPI+.

**Figure 7 nanomaterials-11-01455-f007:**
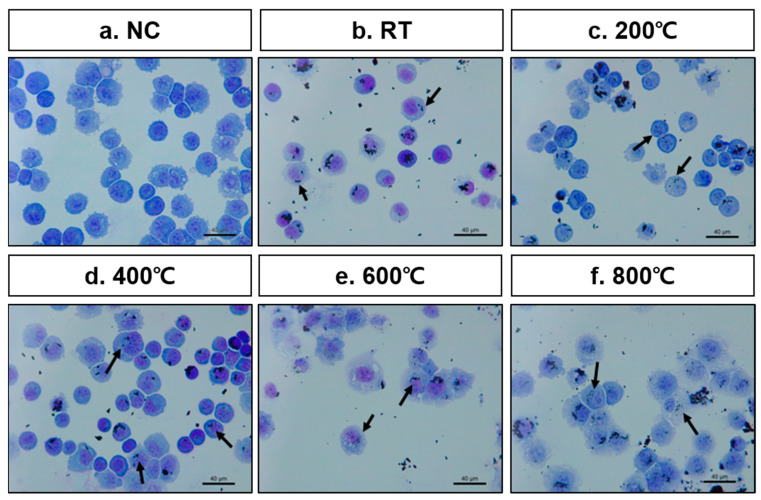
Diff-Quik staining images of (**a**) the naïve control (NC) and (**b–f**) A549 cells stimulated by aBC synthesized at five treatment temperatures. Arrows indicate the internalization of aBC in the A549 cell.

**Figure 8 nanomaterials-11-01455-f008:**
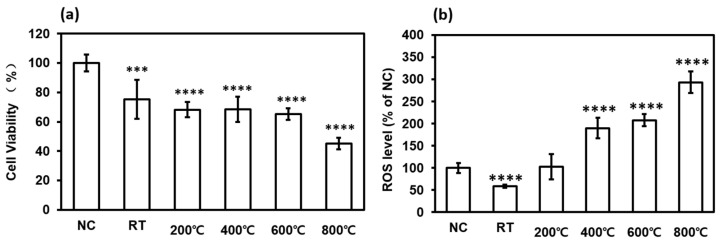
(**a**) Cell viability and (**b**) ROS production in the naïve control (NC) and synthesized aBC-stimulated A549 cells. Data are presented as the mean ± SD (n = 8). *** *p* < 0.01 and **** *p* < 0.001 compared to the NC.

## Data Availability

Data are contained within the article.

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
