# Peer review of "Relationship between Cytotoxicity and Surface Oxidation of Artificial Black Carbon"

_nanomaterials, 2021, doi:10.3390/nano11061455_

Round 1
Reviewer 1 Report
The work of Le et al. can provide a valuable contribution into the understanding of toxic effects of environmental pollutants, but the manuscript requires major revision. My main concern is that the literature background has not been sufficiently covered. Several highly relevant recent articles on the cytotoxicity of both carbon black and soot particles in cultured lung cells have not been cited, such as (i) Lindner et al. Particle Fibre Toxicol. 2017, 14:8, doi 10.1186/s12989-017-0189-1; (ii) Panomsuvan et al. Jap. J. Appl. Phys. 2018, 57, 0102BG, doi 10.7567/JJAP.57.0102BG; or (iii) Jia et al. Environ. Sci. Technol. 2020, 54, 5608-5618, doi 10.1021/acs.est.9b06395. The authors should explain clearly what the novelty of their experimental approach is compared with the previously published works. I also have several concerns about the results of cell-based cytotoxicity studies, which is my area of expertise. First, the cytotoxicity of the studied carbon nanoparticles in A549 cells appears to be very low (cell viability decreases up to ~50% after 48 h treatment with 2 mg/mL particles, line 134 in the text and Figure 8a). By contrast, Hou et al. (Sci. Total Environ. 2020, 698, 134122, doi 10.1016/j.scitotenv.2019.134122) reported that the IC50 value of carbon black nanoparticles in A549 cells was 70 ug/mL in 48 h treatment. A likely reason for the decreased toxicity of carbon nanoparticles in the current work is their aggregation in cell culture medium, which is observed by light microscopy (Figure 7) and is likely to prevent the cellular uptake of nanoparticles. The images shown in Figure 7 cannot be used as a proof of cellular uptake, because internalized particles cannot be distinguished from those adsorbed on the cell membrane. More advanced techniques, such as confocal microscopy (Chen et al. RSC Adv. 2018, 8, 35246, doi 10.1039/c8ra06665e) or TEM (Jia et. al, see above), would be required for this task. In the description of ROS determination assay (lines 153-160), it is unclear whether the cells were treated with nanoparticles before or after loading with DCF-DA, and what treatment conditions were used. Cell treatments conditions should also be indicated in the captions of Figures 7 and 8. Finally, the manuscript requires professional English editing. Some expressions, such as ‘lacking of non-available laboratory BC samples’ (lines 41-42), are not comprehensible.
Reviewer 2 Report
In my opinion, the toxicological characterization of artificial models is difficult to justify for environmental purposes. As the authors state, NIST produce proper samples of BC although with other pollutants. Maybe, a treatment of this samples would lead to the models the authors require for their studies.
Reviewer 3 Report
The manuscript submitted by Yen Thi-Hoang Le and co-workers is focused on the relationships between oxidative degradation of artificially obtained Black Carbon particles and their induced cytotoxicity. The relevance of synthesizing and evaluating the toxic outcome of such particles is argued by the authors through the absence of adequate laboratory samples of Black Carbon.
Overall, the manuscript is systematically, clear and concise written, provides the necessary information to understand the motivation and methods described, is relative sufficiently argued with data, figures and discussions. However, even if claims are largely supported by scientific data provided, there are some suggestions/remarks that should be addressed to the authors to revise and improve their work before an eventual acceptance for publishing in Nanomaterials journal:
- The artificially generated Black Carbon (aBC) nanoparticles are graphite-sourced, with mean sizes of 27.7±3.8 nm (TEM). These sizes are appropriate for modeling the pollution given by incomplete combustion of fossil fuels, but are well under the real-life particles resulted from biomass burn. The authors should better underline the applicability of their simplified model;
- Furthermore, manuscript is also focused on the relationship cytotoxicity / surface (thermal) oxidation of aBC, but oxidative processes seems to be rather poorly highlighted, despite their key role. So, the accuracy level of XPS method has its own limitations especially when a low discrimination within results needs very fine tunes of curves; data extracted from low resolution curves and their fitting have a certain degree of subjectivism even if the authors did their best to be scientifically rigorous. RAMAN is a useful but indirect addition, does not provide accurate conclusions regarding oxidation. FTIR analyzes help to qualitatively approximate the degree of oxidation in a certain extent. In my opinion, these studies should be accompanied by at least one other analysis to determine the oxygen increase and support the XPS data;
- Cytotoxicity did not reside only in the surface oxidation degree. Features that aggressively modify the cytotoxicity of nanoparticles, like surface morphology and porosity are visible changed by thermal treatment (figure 4). A supplementary argument consists in the rather limited concordance between cell viability and ROS production (figure 8). The authors should greatly improve discussions here. Moreover, the main conclusion of the manuscript is: an increase in the presence of oxygen functional groups on the surface of aBC directly affected cell viability and oxidative stress in A549 cells. This statement is not entirely true and is just partially sustained in the submitted form of the manuscript.
In conclusion, the manuscript could not be accepted for publication in the current form. Nevertheless, the manuscript consists in an original work with potential scientific importance and is consistent with the scope and aims of the Nanomaterials journal, so it could be accepted after major revisions made by the authors.

Reviewer 4 Report
The paper of Le et al could be a very interesting study regarding the cytotoxicity of oxidized black carbon nanoparticles, but the toxicological studies are of a very low quality. In the introduction, a presentation regarding the correlation between particle size and the final site of deposition should be done.
In Materials and Methods section, should be presented the dose of nanoparticles used for toxicological studies.
A normal toxicological study should follow the effects of several doses at different time intervals. Only one dose and one time point are not really informative for the comparison between different types of black carbon nanoparticles. This study is completely inappropriate from the toxicological point of view.
Round 2
Reviewer 1 Report
The authors have addressed most of the reviewers’ comments, but the description of biological experiments is still problematic. In particular, the text added in lines 142-148 does not make much sense. First, it should have been added to Section 2.3.3 (Cytotoxicity), rather than to Section 2.3.1. If my interpretation is correct, I would re-write this text as follows:
‘For cytotoxicity experiments, A549 cells were seeded in 96-well plates at the density of 5x104 cells per well and left to attach overnight. The cells were then treated with various concentrations (50-2000 ug mL-1 of aBC in complete medium for 48 h. This was followed by the addition of 3-(4,5-dimethylthiazol-2-thiazyl)-2,5-diphenyl-tetrazolium bromide (MTT) solution to the final concentration of 1.0 mg mL-1. After 3 h incubation, the medium was removed, the formed blue formazan crystals were dissolved in 100 uL per well DMSO, and the absorbance at 570 nm (690 nm background control) was measured using a microplate reader. Low toxicity of aBC was observed in most treatments, with ~75% cells remaining viable after the treatment with the highest dose (2.0 mg mL-1) of aBC for 48 h. These treatment conditions were used for all subsequent experiments.’
I would also like to make a comment that the seeding density of A549 cells was very high (if the 5x104 cells per well number is correct), which probably lead to cells being confluent before the treatment. The best practice for MTT assays is to start with low cell numbers, such as (1-2)x103 cells per well in the case of A549 cells, and let them re-grow for 48-72 h. Using too high number of cells overloads the response of MTT assays, so that the observed cytotoxicity appears lower than it should be, which was probably the case for this work. A good reference for the optimization of MTT assays is Sylvester, P. W. Methods Mol. Biol. 2011, 716, 157-168.
In addition, the text added to line 167 says that the cells were washed with PBS before the DCF-DA treatment. It probably should have said that the cells were treated with aBC (2.0 mg mL-1 for 48 h?), then washed with PBS and loaded with DCF-DA.
Despite these problems, I think that the work provides useful preliminary results on the biological activity of well-characterized combustion products, which deserve to be published.
Reviewer 2 Report
I thank the authors their response. However, I still think that the use of real standards is fundamental in such kind of research.
Author Response
Dear reviewer 3,
We totally respect your opinion in terms of the utilization of real standard particles for toxicity tests. It is better to get specific toxic level or toxic mechanism of specific soot species, however, our objectives for this research are a bit different. We would like to highlight only one property of synthesized aBC which is "chemical surface" which correlates to toxicological responses, therefore, very original particles are more feasible for this research.
Thank you.
Reviewer 3 Report
Journal: Nanomaterials
Manuscript ID: nanomaterials-1210345
Title: Relationship between cytotoxicity and surface oxidation of artificial black carbon
The revised version of the manuscript shows improvements in discussions and clarity.
Nevertheless, there are still flows and author's responses only partially cover the comments addressed. For example, the cytotoxicity of synthesized aBC continues to be correlated with biomass burning aerosols, without considering the difference in sizes. Besides ROS production, the nanoparticulate size is a key factor in the resulting active surface and cytotoxicity. Other factors like nanoparticle’s porosity and topography are also silenced. Or the effect of ROS production is considered and discussed as the single cause of cytotoxicity, which may alters the conclusions of this work.
I agree that this work consists in preliminary, start-up research. However, some discussions, at least theoretically, should be used in my opinion to fix such aspects that are completely omitted, respective to ensure a solid report.
In my opinion, the revised manuscript should be further improved before publication in Nanomaterials journal.
Reviewer 4 Report
The paper of Le et al, could be accepted for publication in this form, but some corrections of English language are necessary.
